# OpenReview forum: "From Judgment to Interference: Early Stopping LLM Harmful Outputs via Streaming Content Monitoring"
_NeurIPS.cc/2025/Conference — NeurIPS 2025 poster_

### Official Review · Reviewer_jUba · 2025-06-30

**Clarity:** 3
**Significance:** 3
**Originality:** 3
**Rating:** 4
**Confidence:** 4

**Summary:**

To achieve the early termination of harmful outputs from large language models (LLMs), this paper proposes a data-and-model method to monitor the output streams and detect the harmfulness of the stream without knowing the complete output. More specifically, it first constructs a dataset called FineHarm for harmfulness detection with both response-level and token-level labels. Then it introduces a streaming content monitor (SCM) and trains it over the FineHarm dataset with a customized loss function. Experimental results show that the trained SCM can achieve comparable performance with the baselines using the full output for detection, while using only partial tokens (18% tokens on average).

**Questions:**

The questions are listed as follows.
1. For the token-level labeling, does the proposed heuristic annotation approach use the delete method in Appendix B.1?
2. What is the relationship between the holistic scorer and the token scorer? Do they share the model architecture or weights?
3. For Equation (4), are there any typos in step 2 and 3? For example, in step 2, should it be $\vee P(f_{hol}(h_{n})) \cdot P( (f_{tok} (h_{i}) )_{(i=1)}^{n})$?
4. What is the stopping rule to get the results in Figure 4? Is delay-k equal to 1?

**Ethical Concerns:**

["NO or VERY MINOR ethics concerns only"]

**Final Justification:**

My questions are all answered with a clear explanation in the rebuttal from the authors. And my major concerns about the OOD performance and the additional baselines have been addressed by the results provided in the rebuttal. I suggest that the authors include these results in the final version. After reading the rebuttal and review from the other reviewers, I keep my original ratings to borderline accept this paper.

**Limitations:**

No. The main concern is on the generalization of SDM. Providing an evaluation on the out-of-distribution test set is helpful to show the generalization.

**Quality:**

3

**Strengths And Weaknesses:**

The strengths of the paper are listed as follows.
1. The paper is well-motivated. The accurate detection of harmful responses from LLM is important for its safety and reducing the latency incurred by the detection in online applications is of great importance for user experience.
2. The authors provide open-source codes and data from this paper, making it easy to follow up and reproduce.
3. The paper is well-written and easy to follow. Figure 1, 2, and 3 clearly illustrate the technical novelty, the necessity of SCM training and the SCM overview of the proposed method, respectively.
4. It is great to see that the proposed method can further improve the implicit safety of LLM by applying the token-level prediction from SCM to the typical safety alignment techniques like DPO. It may improve the impact of the proposed method.

The weaknesses of the paper are listed as follows.
1. The details of the test data for evaluation are unclear. What is its relationship with the training data (FineHarm)? Is there any distribution shift here? It would be better to see the performance of the SCM on the out-of-distribution test data, since the construction of the training data (FineHarm) is not cheap. The poor performance on the out-of-distribution may limit the usage of the proposed data-and-model pipeline.
2. It would be better if the false alarm rate could be shown to see the accuracy of SDM on benign responses.
3. For the evaluation of Table 2, for the results where full detection is better, it would be better to see whether there exists a k value such that the performance is better than full detection. If the k exists, then it can be claimed that the proposed method can be at least as good as the full detection with appropriate k.
4. More justification of baselines should be provided. I think it is important to compare the performance of the proposed method with the existing method mentioned in the related work about LLM Harmful Output Mitigation such as [31, 29, 15, 37] and the ones in Fine-grained Harmful Content Detection. If the related work is not involved in the evaluation, it would be better if the authors could provide explicit justification about the reason to not including them.

---

> ### Author Rebuttal · Authors · 2025-07-29
>
> We are delighted by the reviewer's recognition of the well-motivated focus, the clarity of the writing and figures, as well as the potential impact of FineHarm and SCM. Below, we address the concerns and suggestions in detail:
>
> ---
>
> # 1 Details of test data and OOD testing (W1)
>
> The train-test split follows random split and thus our test is under the i.i.d setting. To test models' capacity under the OOD setting, we further tested SCM (trained on FineHarm) on two existing, non-overlapping benchmarks, Toxic-Chat [1] and ToxiGen [2], and compared the performance with the OOD baselines derived from [1, 2], i.e., HateBERT and ToxDectRoBERTa.
>
> Toxic-Chat:
>
> | Model | Benign Precision | Benign Recall | Benign F1 | Harmful Precision | Harmful Recall | Harmful F1 | Macro F1 |
> |:--:|:--:|:--:|:--:|:--:|:--:|:--:|:--:|
> | HateBERT | 93.36 | 97.06 | 95.17 | 20.57 | 9.94 | 13.41 | 54.29 |
> | ToxDectRoBERTa| 94.06 | 99.62 | 96.76 | 78.31 | 17.96 | 29.21 | 62.99 |
> | SCM 0.5B | 95.03 | 95.64 | 95.33 | 37.95 | 34.81 | 36.31 | 65.82 |
> | SCM 1.5B | 95.11 | 96.89 | 95.99 | 46.35 | 35.08 | 39.94 | 67.96 |
> | SCM 7B | 95.64 | 96.12 | 95.88 | 45.86 | 42.82 | 44.29 | 70.08 |
>
> ToxiGen:
>
> | Model | Benign Precision | Benign Recall | Benign F1 | Harmful Precision | Harmful Recall | Harmful F1 | Macro F1 |
> |:--:|:--:|:--:|:--:|:--:|:--:|:--:|:--:|
> | HateBERT | 75.97 | 83.23 | 79.43 | 55.60 | 44.37 | 49.36 | 64.39 |
> | ToxDectRoBERTa| 73.42 | 87.46 | 79.83 | 55.56 | 33.11 | 41.49 | 60.66 |
> | SCM 0.5B | 80.11 | 66.93 | 72.93 | 48.16 | 64.90 | 55.29 | 64.11 |
> | SCM 1.5B | 82.83 | 77.90 | 80.29 | 58.53 | 65.89 | 61.99 | 71.14 |
> | SCM 7B | 83.77 | 75.24 | 79.27 | 56.95 | 69.21 | 63.48 | 70.88 |
>
> Under the OOD setting:
>
> - SCM has a stronger generalization ability than the baselines in the original papers [2,3].
> - On Toxic-chat, the performance of SCM-7B under OOD testing is comparable to the i.i.d. performance reported in [2] (F1=72.1).
>
> > [1] ToxicChat: Unveiling Hidden Challenges of Toxicity Detection in Real-World User-AI Conversation. EMNLP’23 Findings
> >
> > [2] ToxiGen: A Large-Scale Machine-Generated Dataset for Adversarial and Implicit Hate Speech Detection. ACL’22
>
> # 2 Baseline comparison and supplementary experimental results (W2/W3/W4)
>
> - We added more existing methods, including open-source (HateBERT, ToxDectRoBERTa-Large, and LlamaGuard-3-8B) and closed-source ones (Google Perspective and OpenAI Moderation API).
> - We increased the maximum allowed $k$ from 10 to 50, and the optimal macroF1 of SCM-0.5B improved from 95.64 (when k=4) to 96.08 (when k=13).
>
> The results with more details (MAR/FAR/k) are shown below, which show that the proposed SCM using partial detection can still rival the performance of the added baselines using full detection.
>
> | Model | Partial? | Benign Precision | Benign Recall | Benign F1 | Harmful Precision | Harmful Recall | Harmful F1 | Macro F1 | MAR | FAR | k |
> | :--: | :--: | :--: | :--: | :--: | :--: | :--: | :--: | :--: | :--: | :--: | :--: |
> | HateBERT | ✘ | 95.93 | 94.40 | 95.16 | 93.04 | 94.93 | 93.98 | 94.57 | 5.57 | 5.6 | NA |
> | ToxDectRoBERTa-Large | ✘ | 96.77 | 97.72 | 97.24 | 97.08 | 95.87 | 96.47 | 96.85 | 4.13 | 2.28 | NA |
> | Google Perspective | ✘ | 74.20 | 89.78 | 81.25 | 82.38 | 60.48 | 69.75 | 75.50 | 39.52 | 10.22 | NA |
> | OpenAI Moderation | ✘ | 72.72 | 99.94 | 84.18 | 99.85 | 52.53 | 68.85 | 76.51 | 47.47 | 0.06 | NA |
> | LlamaGuard-3-8B | ✘ | 87.73 | 97.72 | 92.46 | 96.63 | 82.70 | 89.12 | 90.79 | 17.3 | 2.28 | NA |
> | ModernBERT | ✘ | 96.98 | 96.92 | 96.86 | 96.10 | 95.95 | 96.02 | 96.44 | 4.05 | 3.08 | NA |
> | | ✔ | 97.04 | 90.70 | 93.76 | 89.13 | 96.46 | 92.66 | 93.21 | 3.51 | 9.30 | 10 |
> | Qwen 0.5B | ✘ | 96.65 | 96.80 | 96.26 | 95.89 | 94.54 | 95.21 | 95.74 | 5.46 | 3.20 | NA |
> | | ✔ | 92.15 | 87.57 | 89.77 | 85.13 | 90.57 | 87.76 | 88.77 | 9.43 | 12.5 | 5 |
> | **SCM 0.5B** | **✔** | **95.82** | **97.35** | **96.58** | **96.58** | **94.62** | **95.59** | **96.08** | **5.38** | **2.65** | **13** |
> | Qwen 1.5B | ✘ | 98.64 | 98.15 | 98.40 | 97.68 | 98.28 | 97.98 | 98.19 | 1.71 | 1.85 | NA |
> | | ✔ | 98.70 | 79.19 | 87.87 | 78.93 | 98.67 | 87.70 | 87.79 | 1.33 | 20.81 | 10 |
> | **SCM 1.5B** | **✔** | **98.39** | **97.91** | **98.15** | **97.37** | **97.97** | **97.67** | **97.91** | **2.03** | **2.09** | **4** |
> | Qwen 7B | ✘ | 93.87 | 93.29 | 93.58 | 91.57 | 92.28 | 91.93 | 92.75 | 7.72 | 6.71 | NA |
> | | ✔ | 98.09 | 63.18 | 76.85 | 67.87 | 98.44 | 80.34 | 78.60 | 1.56 | 36.82 | 10 |
> | **SCM 7B** | **✔** | **97.78** | **97.72** | **97.75** | **97.12** | **97.19** | **97.16** | **97.45** | **2.81** | **2.28** | **4** |
>
> # 3 Further clarification of the paper's content (Q1/Q2/Q3/Q4)
>
> Thank you for pointing out the unclear expressions, which has been of great help to us. We have revised the relevant expressions in the subsequent version and will provide further clarification below.
>
> > Q1: For the token-level labeling, does the proposed heuristic annotation approach use the delete method in Appendix B.1?
>
> The delete method was an unsuccessful attempt, and the approach we have adopted does not involve any deletion operations. The ultimately adopted steps are described in Section 3.2 of the main text.
>
> > Q2: What is the relationship between the holistic scorer and the token scorer? Do they share the model architecture or weights?
>
> The token scorer and holistic scorer are two linear layers for a hidden_size-to-2 mapping (corresponding to harmful or not) with only a slight difference (the holistic scorer does not set a bias while the token scorer does). They do not share weights.
>
> > Q3: For Equation (4), are there any typos in steps 2 and 3? For example, in step 2, should it be $\vee P\left(f_{h o l}\left(h_{n}\right)\right) \cdot P\left(\left(f_{t o k}\left(h_{i}\right)\right)_{(i=1)}^{n}\right)$?
>
> This equation holds due to the equivalence in propositional logic between implication and a combination of negation and disjunction. An implication $A \Rightarrow B$ is only false in one scenario: when $A$ is true, but $B$ is false. In all other cases, the implication holds.
>
> The truth-value table is as follows, which lists all possible truth values for $A$ and $B$ and compares the two expressions:
>
> | $A$ | $B$ | $A \Rightarrow B$ | $\neg A$ | $\neg A \vee B$ |
> |:--:|:--:|:--:|:--:|:--:|
> | True | True | True | False | True |
> | True | False | False | False | False |
> | False | True | True | True | True |
> | False | False | True | True | True |
>
> Notice that the columns for $A \Rightarrow B$ and $\neg A \vee B$ are identical, indicating the logical equivalence. So we have steps 2 and 3 in Eq. (4).
>
> > Q4: What is the stopping rule to get the results in Figure 4? Is delay-k equal to 1?
>
> The termination position depends on the $k$ value when the best performance in Table 2 is achieved. It can also be directly seen from the results for the second reply.
>
> ---
>
> Thank you again for your insightful feedback.
>
> We believe these revisions will strengthen the paper's clarity. Please let us know if further clarifications are needed.
>
> Best regards

---

> > ### Comment · Reviewer_jUba · 2025-08-07
> >
> > Thank you for the detailed rebuttal from the authors. I have read all of the content and believe that the rebuttal addressed most of my concerns and questions. Therefore, I tend to keep my original rating to accept this paper.

---

> > > ### Author Response · Authors · 2025-08-07
> > >
> > > Your insightful comments are crucial for enhancing our work, and we sincerely appreciate the valuable time you have devoted to reviewing.
> > >
> > > Best regards

---

### Official Review · Reviewer_s3cF · 2025-07-02

**Clarity:** 3
**Significance:** 3
**Originality:** 2
**Rating:** 4
**Confidence:** 4

**Summary:**

This paper focuses on a critical problem: existing moderation methods rely heavily on full-response evaluation, making them unsuitable for streaming generation scenarios. The main contributions are two-fold: (1) the construction of FineHarm, a token-level fine-grained harmfulness annotation dataset; and (2) the proposal of SCM, a streaming content moderator that natively supports partial detection. By leveraging fine-grained data and a hierarchical training strategy, SCM achieves comparable performance to full-response detection by inspecting only the first 18% of the response, significantly enhancing the real-time safety of LLM systems.

**Questions:**

Although SCM runs in parallel with LLMs in streaming settings, several practical concerns remain: (1) Can SCM keep up with the LLM, i.e., is the token-level prediction completed before the next token is generated? (2) If SCM is slower than the LLM's generation speed, its per-token inference cost may outweigh the efficiency gains—since full-response detection requires only a single pass.

**Ethical Concerns:**

["NO or VERY MINOR ethics concerns only"]

**Final Justification:**

Most of my concerns were addressed. The discussion of the inference-latency implications of partial detection and the OOD test results appear promising.

**Limitations:**

yes

**Quality:**

3

**Strengths And Weaknesses:**

## Strengths

1. Leveraging partial detection for safety moderation in streaming LLM generation is a highly valuable research direction.
2. The token-level fine-grained annotation dataset proposed in this work facilitates the development of partial detection techniques for LLM safety.
3. The proposed Streaming Content Monitor (SCM) is simple yet highly effective, as demonstrated by experimental results.
4. SCM can also be used to construct token-level preference datasets, and experiments show that token-level DPO outperforms response-level DPO in enhancing safety.

##Weaknesses

1. The proposed SCM method shares similarities with Speculative Decoding [1,2], which relies on generating draft tokens followed by parallel verification. Therefore, it is important to clearly explain the key differences between SCM and Speculative Decoding in the related work section.
2. The main methodological innovation of SCM lies in its Hierarchical Consistency-Aware Learning, which jointly optimizes three loss functions. However, the experimental section only includes an ablation study on the logical consistency loss. A comprehensive ablation analysis of the holistic loss design would strengthen the empirical validation.
3. The main evaluation dataset used in the experiments appears to be under-specified. It remains unclear whether it shares the same source or distribution as the training data. In the context of safety detection, out-of-distribution generalization is crucial, as jailbreak attacks are highly diverse and unpredictable. Clarifying the train/test dataset properties and evaluating generalization would greatly enhance the credibility of the results.
4. The baselines are limited. Comparing a task-specific fine-tuned model with general-purpose LLMs is not entirely fair. Key safety models like LlamaGuard are also missing from the evaluation.

[1] Fast Inference from Transformers via Speculative Decoding

[2] AdaEDL: Early Draft Stopping for Speculative Decoding of Large Language Models via an Entropy-based Lower Bound on Token Acceptance Probability

---

> ### Author Rebuttal · Authors · 2025-07-30
>
> Thank you for highlighting the significance of the problem we addressed and the value of the FineHarm dataset and SCM to the community. Below, we address your concerns and suggestions in detail:
>
> ---
>
> # 1 Differences between SCM and Speculative Decoding (W1)
>
> Thank you for providing an insightful perspective. We agree that SCM shares similarities with speculative decoding in terms of the partial context they are applied and the token-level evaluation with the assistance of a (commonly smaller) model. However, SCM and speculative decoding are also very distinct, which we’d like to clarify through three core dimensions:
>
> - **Different motivation.** The design of speculative decoding is to accelerate the generation process, improving the inference efficiency. In contrast, the core goal of SCM is to conduct a real-time assessment of potential risks in the output stream, and it does not influence the token decoding of the monitored LLM.
> - **Different functional positionings of the external model.** In speculative decoding, **the introduced small LM is used to draft tokens**, and the verifier is the large model itself; however, in our method, the monitored model (regardless of its size) will draft tokens, and **the small LM we introduce (i.e., SCM) is used for verification** (the harmfulness).
> - **Different standpoints for cost optimization.** Speculative decoding optimizes the **single-token** generation stage by replacing the inference of the large LM with the smaller one, while SCM considers the costs in the **overall generation** perspective (via stopping the output early by observing fewer tokens, thus eliminating the unnecessary cost caused by detection after complete outputs).
>
> Due to the differences in motivation, the roles of models, and the standpoints for cost optimization, we believe that the design of SCM is deeply rooted in the comprehension of harmful output monitoring scenarios and is non-trivial even if speculative decoding exists. We plan to share such a technical discussion with readers and cite related works in the revised version.
>
> # 2 Ablation of holistic loss design (W2)
>
> After removing the Holistic Scorer, the detection performance drops, possibly due to the absence of future knowledge injection during learning that reflects the harmfulness to the entire response.
>
> | Model | Benign Precision | Benign Recall | Benign F1 | Harmful Precision | Harmful Recall | Harmful F1 | Macro F1 |
> |:--:|:--:|:--:|:--:|:--:|:--:|:--:|:--:|
> | SCM 0.5b | 96.18 | 96.12 | 96.15 | 95.09 | 95.17 | 95.13 | 95.64 |
> | w/o logic | 92.51 | 98.89 | 95.60 | 98.46 | 89.87 | 93.97 | 94.78 |
> | w/o holistic | 89.36 | 99.32 | 94.08 | 99.00 | 85.04 | 91.49 | 92.78 |
> | SCM 1.5b | 98.39 | 97.91 | 98.15 | 97.37 | 97.97 | 97.67 | 97.91 |
> | w/o logic | 93.30 | 93.53 | 93.42 | 91.79 | 91.50 | 91.65 | 92.53 |
> | w/o holistic | 90.93 | 99.88 | 94.88 | 99.82 | 86.52 | 92.69 | 93.79 |
> | SCM 7b | 97.78 | 97.72 | 97.75 | 97.12 | 97.19 | 97.16 | 97.45 |
> | w/o logic | 94.27 | 99.26 | 96.70 | 99.00 | 92.36 | 95.56 | 96.13 |
> | w/o holistic | 87.04 | 98.83 | 92.56 | 98.21 | 81.37 | 89.00 | 90.78 |
>
> # 3 Out-of-distribution (OOD) testing (W3)
>
> Since the test data shares the same source or distribution as the training data, we further tested SCM (trained on our FineHarm) on two existing, non-overlapping benchmarks, Toxic-Chat [1] and ToxiGen [2], and compared the performance with the OOD baselines derived from [1, 2], i.e., HateBERT and ToxDectRoBERTa.
>
> Toxic-Chat:
>
> | Model | Benign Precision | Benign Recall | Benign F1 | Harmful Precision | Harmful Recall | Harmful F1 | Macro F1 |
> |:--:|:--:|:--:|:--:|:--:|:--:|:--:|:--:|
> | HateBERT | 93.36 | 97.06 | 95.17 | 20.57 | 9.94 | 13.41 | 54.29 |
> | ToxDectRoBERTa| 94.06 | 99.62 | 96.76 | 78.31 | 17.96 | 29.21 | 62.99 |
> | SCM 0.5b | 95.03 | 95.64 | 95.33 | 37.95 | 34.81 | 36.31 | 65.82 |
> | SCM 1.5b | 95.11 | 96.89 | 95.99 | 46.35 | 35.08 | 39.94 | 67.96 |
> | SCM 7b | 95.64 | 96.12 | 95.88 | 45.86 | 42.82 | 44.29 | 70.08 |
>
> ToxiGen:
>
> | Model | Benign Precision | Benign Recall | Benign F1 | Harmful Precision | Harmful Recall | Harmful F1 | Macro F1 |
> |:--:|:--:|:--:|:--:|:--:|:--:|:--:|:--:|
> | HateBERT | 75.97 | 83.23 | 79.43 | 55.60 | 44.37 | 49.36 | 64.39 |
> | ToxDectRoBERTa| 73.42 | 87.46 | 79.83 | 55.56 | 33.11 | 41.49 | 60.66 |
> | SCM 0.5b | 80.11 | 66.93 | 72.93 | 48.16 | 64.90 | 55.29 | 64.11 |
> | SCM 1.5b | 82.83 | 77.90 | 80.29 | 58.53 | 65.89 | 61.99 | 71.14 |
> | SCM 7b | 83.77 | 75.24 | 79.27 | 56.95 | 69.21 | 63.48 | 70.88 |
>
> Under the same OOD setting:
>
> - SCM has a stronger generalization ability than the baselines mentioned in their papers.
> - On the Toxic-chat dataset, the performance of SCM-7B under out-of-distribution testing is comparable to the in-distribution testing value mentioned in the paper (F1=72.1).
>
> > [1] ToxicChat: Unveiling Hidden Challenges of Toxicity Detection in Real-World User-AI Conversation. EMNLP’23 Findings
> >
> > [2] ToxiGen: A Large-Scale Machine-Generated Dataset for Adversarial and Implicit Hate Speech Detection. ACL’22
> >
>
> # 4 Baseline comparison (W4)
>
> We added more existing methods, including open-source (HateBERT, ToxDectRoBERTa-Large, and LlamaGuard-3-8B) and closed-source ones (Google Perspective and OpenAI Moderation API). The results are shown below, which show that the proposed SCM using partial detection can still rival the performance of the supplemented baselines using full detection.
>
> | Model | Partial? | Benign Precision | Benign Recall | Benign F1 | Harmful Precision | Harmful Recall | Harmful F1 | Macro F1 |
> | :--: | :--: | :--: | :--: | :--: | :--: | :--: | :--: | :--: |
> | HateBERT | ✘ | 95.93 | 94.40 | 95.16 | 93.04 | 94.93 | 93.98 | 94.57 |
> | ToxDectRoBERTa-Large | ✘ | 96.77 | 97.72 | 97.24 | 97.08 | 95.87 | 96.47 | 96.85 |
> | Google Perspective | ✘ | 74.20 | 89.78 | 81.25 | 82.38 | 60.48 | 69.75 | 75.50 |
> | OpenAI Moderation | ✘ | 72.72 | 99.94 | 84.18 | 99.85 | 52.53 | 68.85 | 76.51 |
> | LlamaGuard-3-8B | ✘ | 87.73 | 97.72 | 92.46 | 96.63 | 82.70 | 89.12 | 90.79 |
> | ModernBERT | ✘ | 96.98 | 96.92 | 96.86 | 96.10 | 95.95 | 96.02 | 96.44 |
> | | ✔ | 97.04 | 90.70 | 93.76 | 89.13 | 96.46 | 92.66 | 93.21 |
> | Qwen 0.5b | ✘ | 96.65 | 96.80 | 96.26 | 95.89 | 94.54 | 95.21 | 95.74 |
> | | ✔ | 92.15 | 87.57 | 89.77 | 85.13 | 90.57 | 87.76 | 88.77 |
> | **SCM 0.5b** | **✔** | **96.18** | **96.12** | **96.15** | **95.09** | **95.17** | **95.13** | **95.64** |
> | Qwen 1.5b | ✘ | 98.64 | 98.15 | 98.40 | 97.68 | 98.28 | 97.98 | 98.19 |
> | | ✔ | 98.70 | 79.19 | 87.87 | 78.93 | 98.67 | 87.70 | 87.79 |
> | **SCM 1.5b** | **✔** | **98.39** | **97.91** | **98.15** | **97.37** | **97.97** | **97.67** | **97.91** |
> | Qwen 7b | ✘ | 93.87 | 93.29 | 93.58 | 91.57 | 92.28 | 91.93 | 92.75 |
> | | ✔ | 98.09 | 63.18 | 76.85 | 67.87 | 98.44 | 80.34 | 78.60 |
> | **SCM 7b** | **✔** | **97.78** | **97.72** | **97.75** | **97.12** | **97.19** | **97.16** | **97.45** |
>
> # 5 Latency of detection (Q1/Q2)
>
> Let $ t_{gen} $ be the generation delay, $ t_{det} $ be the detection delay, and $ T $ be the length of the generated sequence. If termination occurs at position $ k \leq T $ due to partial detection, then:
>
> 1. For synchronous generation and detection (SCM), the delay for a single token is $ \max(t_{gen}, t_{det}) $. The time taken until termination is $ k \times \max(t_{gen}, t_{det}) $. If $k$ follows uniform distribution, the average time is $\mathbb{E}(t)=\frac{1}{T}\sum_{k=1}^T[k \times \max(t_{gen}, t_{det})] = \frac{T+1}{2} \times \max(t_{gen}, t_{det}) $. [Recalling our empirical results, the distribution peak would be smaller than the midpoint, and thus the avg time would be lower.]
>
> 2. For detection after all generations are completed (Full detection), the total time taken is $ t=T \times t_{gen} + t_{det} $.
>
> We tested the latency under the same setups:
>
> - Without using any acceleration methods.
> - With the test sequence length increasing from 50 to 100.
> - Each test sequence is repeated 10 times to report the min/max/avg latency.
>
> Generation latency ($t_{gen}$) of Qwen:
>
> | Scale | Min (ms) | Max (ms) | Avg (ms) |
> |:--:|:--:|:--:|:--:|
> | 0.5B | 0.23 | 0.46 | 0.32 |
> | 1.5B | 0.28 | 0.53 | 0.38 |
> | 7B | 1,07 | 1.51 | 1.25 |
>
> Detection latency ($t_{net}$) of SCM:
>
> | Scale | Min (ms) | Max (ms) | Avg (ms) |
> |:--:|:--:|:--:|:--:|
> | 0.5B | 0.20 | 0.42 | 0.28 |
> | 1.5B | 0.25 | 0.47 | 0.34 |
> | 7B | 1.05 | 1.47 | 1.22 |
>
> Notice that $t_{gen}>t_{det}$ under the same scale, so the average time of SCM is $\frac{T+1}{2}\times t_{gen}$ and the time of full detection is $T \times t_{gen} + t_{det}\approx T\times t_{gen}$. SCM is faster since $\frac{T+1}{2}<T$ for any $T>1$. If the generation model is larger, the condition $t_{gen}>t_{det}$ still holds. [We omit the situation where the detection model is larger than the generation model, as it is rare in reality.]
>
> Detecting the previous token during the process of generating the next token does not introduce additional time consumption, which means that **only a 1-token delay is needed to ensure that the streaming monitor runs smoothly in parallel and barely affects the user experience.**
>
> ---
>
> Thank you again for your insightful feedback.
>
> We believe these revisions will strengthen the paper's clarity. Please let us know if further clarifications are needed.
>
> Best regards

---

> > ### Comment · Reviewer_s3cF · 2025-08-06
> >
> > Thank you very much for your detailed response. The additional experimental results you provided have significantly enhanced my understanding of the paper, particularly the supplementary analysis regarding the inference latency introduced by partial detection, which I find highly valuable. Moreover, in the “Out-of-distribution (OOD) testing, W3)” section, you acknowledged that the initial experimental setup involved training and testing data from the same distribution. While this experiment alone would be insufficient to fully validate the effectiveness of the SCM method, I appreciate that you supplemented your response with OOD testing results to address this concern.
> > Overall, your response has addressed most of my concerns, and I am inclined to raise my score to 4.

---

> > > ### Author Response · Authors · 2025-08-07
> > >
> > > Thank you once again for your effort. Your detailed review and constructive suggestions have greatly contributed to refining our manuscript.
> > >
> > > We would be truly grateful if you could consider submitting the final rating.
> > >
> > > Best regards

---

### Official Review · Reviewer_1hkT · 2025-07-03

**Clarity:** 3
**Significance:** 3
**Originality:** 3
**Rating:** 4
**Confidence:** 3

**Summary:**

Current LLM moderation paradigms are primarily executed in a post-hoc manner, i.e., they determine the harmfulness of LLM responses after receiving the complete output. This paper introduces  FineHarm, a dataset consisting of 29K prompt-response pairs with fine-grained token-level annotations, that facilitates work on partial detection in LLM output safety assessment. Consequently, the paper proposes the "Streaming Content Monitor" - a method capable of partial detection of harmfulness in the LLM output. SCM fetches the latest token at each step of the autoregressive LLM output generation and provides a harmfulness judgment. When the number of tokens that are judged as harmful reaches a preset threshold, the LLM output is considered harmful and terminated midway.

**Questions:**

Q1: The authors state that no prior partial detectors exist. Thus, as baselines, they apply full-detection models in a stepwise fashion. Expectedly, such baselines underperform, due to the training–inference gap problem (partial vs. full context). I wonder if other reasonable baselines or ablations can be explored. For example, training a token-level safety moderator without the consistency loss on a dataset where each token is naively labelled as harmful/not based on whether the full response is harmful. This would quantify the effect that fine-grained (and not naive) token labels have on the method's performance.

Q2: Could you please elaborate on the computational overhead introduced by this method of safety assessment?

**Ethical Concerns:**

["NO or VERY MINOR ethics concerns only"]

**Final Justification:**

My concerns about the limited comparison with baselines and OOD generalization were also raised by other Reviewers. I believe the rebuttal provides sufficient empirical evidence to address these points. I strongly suggest the authors include these results in the camera-ready version if the paper is accepted.

Unless any critical issue that I may have missed comes up later during the discussion, I’m now leaning towards acceptance. I have increased my score to *borderline accept*.

**Limitations:**

The authors adequately addressed the limitations.

**Paper Formatting Concerns:**

No concerns.

**Quality:**

3

**Strengths And Weaknesses:**

### Strengths
S1: Important problem and good motivation. This work practically addresses a timely safety issue.

S2: Dataset contribution. This work contributes FineHarm - a dataset of LLM outputs with fine-grained, token-level harmfulness labels, which constitutes a valuable resource for the community.

S3: Strong empirical results.  SCM achieves impressive results. While seeing only the beginning of outputs in many cases, SCM reaches comparable performance to a full-sequence approach.

### Weaknesses

W1: Reliance on heuristic annotations. My concern is the quality of the token-level labels in the  FineHarm dataset. According to my understanding, the token labels are generated by algorithmic means, and no human annotation was involved. Thus, this approach may miss subtleties, e.g. some benign tokens might be labelled as harmful just because they are in a sentence flagged as harmful. I note that the authors themselves acknowledge this limitation.

W2: Somewhat incremental method.  The idea of early-stop moderation is not entirely new. Solutions mentioned in the paper, e.g. ProtectAI or GuardrailsAI, have already considered scanning the LLM output stream as it is generated. The main technical contribution of the SCM method is adapting a classifier to operate on partial token sequences. This approach is essentially a form of multi-task or hierarchical learning enabled by training on a dataset with fine-grained labels.

W3: The computational overhead introduced by this safety moderation method is not fully discussed.

W4: In Figure 1: "Existing ones [harmful content detectors] take the complete output of LLMs as input, which leaks harmful
outputs to users before the withdrawal (if outputting immediately) or causes additional service delay (if outputting
after detection).". This seems like an overstatement since the paper itself mentions existing solutions that allow for midway safety assessment.

W4: My understanding is that the goal of Table 2 is to provide a comparison of partial and full detection, showcasing the benefits of the approach leveraged by SMC. However, it seems that there is no comparison with other existing dedicated safety assessment models (even not intended for partial detection).

W5: It seems that there is no evaluation on how SMC generalises to other datasets beyond FineHarm. An evaluation (and comparison) of an SMC trained on FineHarm on another dataset, e.g. ToxiChat, would strengthen the work.

---

> ### Author Rebuttal · Authors · 2025-07-30
>
> Thank you for highlighting the significance of the problem we addressed, the value of the FineHarm dataset to the community, and the strong empirical results achieved by SCM. Below, we address your concerns and suggestions in detail:
>
> ---
>
> # 1 Annotation issue (W1)
>
> Thank you for raising this question, which gives us a chance to further elaborate on the motivation behind the heuristic annotation.
>
> Due to the absence of fine-grained labels, it is hard to train harmful output moderators with incomplete outputs. Our annotation aims to find a cost-acceptable way to provide a reasonably **good starting point** for subsequent models’ learning, even if it may not be as accurate as the expensive human labeling. After testing and giving up the LLM-based deleting approach, we ultimately turned to the relatively conventional way relying on the POS prior, which was then experimentally shown to be useful. The potential noise from heuristic annotations was then tolerated by the proper coordination of data and learning strategies in SCM’s training. As shown in Fig. 5, SCM finally identified harmful words with a reasonable combination of prior knowledge from the annotations and its own learned experience.
>
> By intentionally relaxing the quality requirement of annotations, SCM exemplifies a data-and-model solution that balances the performance and the annotation cost. Among them, the heuristic annotations play a key (though imperfect) role.
>
> # 2 Key contribution (W2)
>
> We sincerely believe that the key contribution is not merely an application of multi-task learning with fine-grained labels.
>
> - For the idea, though ProtectAI and GuardrialsAI support streaming safety monitoring, their implementations are not intended to pursue an early-stopping in nature. Our work identifies the importance of early-stop moderation, reviews existing possible solutions (including ProtectAI’s and GuardrialsAI’s), and formulates the early moderation with actionable settings (i.e., the Delay-k notations).
>
> - For the techniques, we provide a data-and-model solution that tackles the remaining challenges to building such a moderator, including the cheap acquisition of fine-grained token-level annotations and the more effective adaptation of the moderator to the incomplete semantics, resulting in FineHarm and SCM as we introduced. The solution considers the multiple factors (annotation costs, compatibility, etc.) for real-world applications, which we believe is not too straightforward and mature to be a less technical contribution. As we mentioned in the discussions, we hope that SCM could be an inspiration for real-world attempts.
>
> # 3 Computational overhead (W3/Q2)
>
> ## 3.1 FLOPS
>
> We use the `fvcore library` to calculate the number of floating-point operations (FLOPS) required for the model to infer a token. The FLOPS required by the SCM Detection Model is **less than** that required by the Qwen Generation Model.
>
> | Scale | Qwen Generation (10^9 FLOPS) | SCM Detection (10^9 FLOPS) |
> |:--:|:--:|:--:|
> | 0.5B | 0.4940 | 0.3578 |
> | 1.5B | 1.5436 | 1.3102 |
> | 7B | 7.0703 | 6.5253 |
>
> ## 3.2 Latency
>
> Let $ t_{gen} $ be the generation delay, $ t_{det} $ be the detection delay, and $ T $ be the length of the generated sequence. If termination occurs at position $ k \leq T $ due to partial detection, then:
>
> 1. For synchronous generation and detection (SCM), the delay for a single token is $ \max(t_{gen}, t_{det}) $. The time taken until termination is $ k \times \max(t_{gen}, t_{det}) $. If $k$ follows uniform distribution, the average time is $\mathbb{E}(t)=\frac{1}{T}\sum_{k=1}^T[k \times \max(t_{gen}, t_{det})] = \frac{T+1}{2} \times \max(t_{gen}, t_{det}) $. [Recalling our empirical results, the distribution peak would be smaller than the midpoint, and thus the avg time would be lower.]
>
> 2. For detection after all generations are completed (Full detection), the total time taken is $ t=T \times t_{gen} + t_{det} $.
>
> We tested the latency under the same setups (without using any acceleration methods, with the test sequence length increasing from 50 to 100, and each test sequence is repeated 10 times to report the min/max/avg latency).
>
> Generation latency ($t_{gen}$) of Qwen:
>
> | Scale | Min (ms) | Max (ms) | Avg (ms) |
> |:--:|:--:|:--:|:--:|
> | 0.5B | 0.23 | 0.46 | 0.32 |
> | 1.5B | 0.28 | 0.53 | 0.38 |
> | 7B | 1,07 | 1.51 | 1.25 |
>
> Detection latency ($t_{net}$) of SCM:
>
> | Scale | Min (ms) | Max (ms) | Avg (ms) |
> |:--:|:--:|:--:|:--:|
> | 0.5B | 0.20 | 0.42 | 0.28 |
> | 1.5B | 0.25 | 0.47 | 0.34 |
> | 7B | 1.05 | 1.47 | 1.22 |
>
> Notice that $t_{gen}>t_{det}$ under the same scale, so the average time of SCM is $\frac{T+1}{2}\times t_{gen}$ and the time of full detection is $T \times t_{gen} + t_{det}\approx T\times t_{gen}$. SCM is faster since $\frac{T+1}{2}<T$ for any $T>1$. If the generation model is larger, the condition $t_{gen}>t_{det}$ still holds. [We omit the situation where the detection model is larger than the generation model, as it is rare in reality.]
>
> Detecting the previous token during the process of generating the next token does not introduce additional time consumption, which means that **only a 1-token delay is needed to ensure that the streaming monitor runs smoothly in parallel and barely affects the user experience.**
>
> # 4 Baseline comparison (W4)
>
> We added more existing methods, including open-source (HateBERT, ToxDectRoBERTa-Large, and LlamaGuard-3-8B) and closed-source ones (Google Perspective and OpenAI Moderation API). The results are shown below, which show that the proposed SCM using partial detection can still rival the performance of the supplemented baselines using full detection.
>
> | Model | Partial? | Benign Precision | Benign Recall | Benign F1 | Harmful Precision | Harmful Recall | Harmful F1 | Macro F1 |
> | :--: | :--: | :--: | :--: | :--: | :--: | :--: | :--: | :--: |
> | HateBERT | ✘ | 95.93 | 94.40 | 95.16 | 93.04 | 94.93 | 93.98 | 94.57 |
> | ToxDectRoBERTa-Large | ✘ | 96.77 | 97.72 | 97.24 | 97.08 | 95.87 | 96.47 | 96.85 |
> | Google Perspective | ✘ | 74.20 | 89.78 | 81.25 | 82.38 | 60.48 | 69.75 | 75.50 |
> | OpenAI Moderation | ✘ | 72.72 | 99.94 | 84.18 | 99.85 | 52.53 | 68.85 | 76.51 |
> | LlamaGuard-3-8B | ✘ | 87.73 | 97.72 | 92.46 | 96.63 | 82.70 | 89.12 | 90.79 |
> | ModernBERT | ✘ | 96.98 | 96.92 | 96.86 | 96.10 | 95.95 | 96.02 | 96.44 |
> | | ✔ | 97.04 | 90.70 | 93.76 | 89.13 | 96.46 | 92.66 | 93.21 |
> | SCM 0.5b | ✔ | 96.18 | 96.12 | 96.15 | 95.09 | 95.17 | 95.13 | 95.64 |
> | SCM 1.5b | ✔ | 98.39 | 97.91 | 98.15 | 97.37 | 97.97 | 97.67 | 97.91 |
> | SCM 7b | ✔ | 97.78 | 97.72 | 97.75 | 97.12 | 97.19 | 97.16 | 97.45 |
>
> # 5 Out-of-distribution (OOD) testing (W5)
>
> For the OOD setting, we tested SCM (trained on our FineHarm) on two existing, non-overlapping benchmarks, Toxic-Chat (EMNLP’23 Findings) and ToxiGen (ACL’22), and compared the performance with the OOD baselines derived from them, i.e., HateBERT and ToxDectRoBERTa.
>
> Toxic-Chat:
>
> | Model | Benign Precision | Benign Recall | Benign F1 | Harmful Precision | Harmful Recall | Harmful F1 | Macro F1 |
> |:--:|:--:|:--:|:--:|:--:|:--:|:--:|:--:|
> | HateBERT | 93.36 | 97.06 | 95.17 | 20.57 | 9.94 | 13.41 | 54.29 |
> | ToxDectRoBERTa| 94.06 | 99.62 | 96.76 | 78.31 | 17.96 | 29.21 | 62.99 |
> | SCM 0.5b | 95.03 | 95.64 | 95.33 | 37.95 | 34.81 | 36.31 | 65.82 |
> | SCM 1.5b | 95.11 | 96.89 | 95.99 | 46.35 | 35.08 | 39.94 | 67.96 |
> | SCM 7b | 95.64 | 96.12 | 95.88 | 45.86 | 42.82 | 44.29 | 70.08 |
>
> ToxiGen:
>
> | Model | Benign Precision | Benign Recall | Benign F1 | Harmful Precision | Harmful Recall | Harmful F1 | Macro F1 |
> |:--:|:--:|:--:|:--:|:--:|:--:|:--:|:--:|
> | HateBERT | 75.97 | 83.23 | 79.43 | 55.60 | 44.37 | 49.36 | 64.39 |
> | ToxDectRoBERTa| 73.42 | 87.46 | 79.83 | 55.56 | 33.11 | 41.49 | 60.66 |
> | SCM 0.5b | 80.11 | 66.93 | 72.93 | 48.16 | 64.90 | 55.29 | 64.11 |
> | SCM 1.5b | 82.83 | 77.90 | 80.29 | 58.53 | 65.89 | 61.99 | 71.14 |
> | SCM 7b | 83.77 | 75.24 | 79.27 | 56.95 | 69.21 | 63.48 | 70.88 |
>
> Under the same OOD setting:
>
> - SCM has a stronger generalization ability than the baselines mentioned in their papers.
> - On the Toxic-chat dataset, the performance of SCM-7B under out-of-distribution testing is comparable to the in-distribution testing value mentioned in the paper (F1=72.1).
>
> # 6 Annotation ablation (Q1)
>
> We trained a content moderation model (CLF below) using QwenForTokenClassification, which was trained with naively labeled tokens: Each token is labeled as "harmful" or "not harmful" based on whether the response-level label is "harmful". Due to the space limit, we report the results at a 1.5B scale. **This shows that such a naive labeling method introduces more noise, leading to a decline in model performance.**
>
> | Model | Benign Precision | Benign Recall | Benign F1 | Harmful Precision | Harmful Recall | Harmful F1 | Macro F1 |
> | :--: | :--: | :--: | :--: | :--: | :--: | :--: | :---: |
> | ModernBERT-Partial | 97.04 | 90.70 | 93.76 | 89.13 | 96.46 | 92.66 | 93.21 |
> | CLF | 97.39 | 94.33 | 95.84 | 93.10 | 96.80 | 94.29 | 95.38 |
> | SCM | 98.39 | 97.91 | 98.15 | 97.37 | 97.97 | 97.67 | 97.91 |
>
> We see similar findings at the other two scales. We will include the complete results in the revised version.
>
> ---
>
> Thank you again for your insightful feedback, which has helped us greatly improve the quality of the paper.
>
> We believe these revisions will strengthen the paper's clarity. Please let us know if any further clarification is needed.
>
> Best regards

---

### Official Review · Reviewer_JZGk · 2025-07-03

**Clarity:** 2
**Significance:** 4
**Originality:** 3
**Rating:** 4
**Confidence:** 3

**Summary:**

The paper focuses on the detection of harmful content during the streaming output of LLMs without knowing the full response. It first introduces FineHarm, a dataset with token-level safety annotations for each response, then proposes the streaming content monitor to detect harmful content during streaming output. Specifically, the training of SCM applies the token loss and the holistic loss to optimize the safety prediction of tokens and response, respectively, and the logic loss to constrain the logical consistency. The paper then introduces delay-k partial detection, which stops streaming outputs when k harmful tokens are detected. Experimental results demonstrate that SCM surpasses the baseline in partial detection. Meanwhile, it shows additional applications in the safety alignment of LLMs with token-level DPO.

**Questions:**

It seems that only the sensitivity of $\alpha$ in Figure 7. Does $\beta$ have a significant impact on the performance?

**Ethical Concerns:**

["NO or VERY MINOR ethics concerns only"]

**Limitations:**

yes

**Quality:**

3

**Strengths And Weaknesses:**

### **Strength**
1. The paper addresses an important and timely problem: detecting harmful content during LLM streaming outputs as early as possible.
2. The paper proposes a novel data-and-model solution including both the data annotation and the consistency-aware learning, which could provide insight for the harmful content detection.
3. The experiments illustrate strong results on the partial detection of harmful content, which shows its potential in real-world applications.
### **Weakness**
1. The baseline could be improved. While there could be a few existing works exactly addressing partial detection, it is still suggested to compare its performance with other harmful content detection methods, such as LlamaGuard, for both full detection and partial detection.
2. The generalizability of the proposed method is not explored. It seems that the test data of FineHarm is totally sampled from the uncensored Llama-3.1 with WildJailbreak, considering harmful content from various LLMs and benchmarks in the evaluation could help to better understand the practicability of the method.

---

> ### Author Rebuttal · Authors · 2025-07-29
>
> We would like to thank the reviewers for acknowledging the contributions of our work in terms of motivation, data-and-model solution, and experiments, as well as for providing detailed evaluations and constructive feedback. Below, we address the concerns and suggestions in detail:
>
> ---
>
> # 1 Baseline comparison (W1)
>
> We added more existing methods, including open-source (HateBERT, ToxDectRoBERTa-Large, and LlamaGuard-3-8B) and closed-source ones (Google Perspective and OpenAI Moderation API). The results are shown below, which show that the proposed SCM using partial detection can still rival the performance of the supplemented baselines using full detection.
>
> | Model | Partial? | Benign Precision | Benign Recall | Benign F1 | Harmful Precision | Harmful Recall | Harmful F1 | Macro F1 |
> | :--: | :--: | :--: | :--: | :--: | :--: | :--: | :--: | :--: |
> | HateBERT | ✘ | 95.93 | 94.40 | 95.16 | 93.04 | 94.93 | 93.98 | 94.57 |
> | ToxDectRoBERTa-Large | ✘ | 96.77 | 97.72 | 97.24 | 97.08 | 95.87 | 96.47 | 96.85 |
> | Google Perspective | ✘ | 74.20 | 89.78 | 81.25 | 82.38 | 60.48 | 69.75 | 75.50 |
> | OpenAI Moderation | ✘ | 72.72 | 99.94 | 84.18 | 99.85 | 52.53 | 68.85 | 76.51 |
> | LlamaGuard-3-8B | ✘ | 87.73 | 97.72 | 92.46 | 96.63 | 82.70 | 89.12 | 90.79 |
> | ModernBERT | ✘ | 96.98 | 96.92 | 96.86 | 96.10 | 95.95 | 96.02 | 96.44 |
> | | ✔ | 97.04 | 90.70 | 93.76 | 89.13 | 96.46 | 92.66 | 93.21 |
> | Qwen 0.5b | ✘ | 96.65 | 96.80 | 96.26 | 95.89 | 94.54 | 95.21 | 95.74 |
> | | ✔ | 92.15 | 87.57 | 89.77 | 85.13 | 90.57 | 87.76 | 88.77 |
> | **SCM 0.5b** | **✔** | **96.18** | **96.12** | **96.15** | **95.09** | **95.17** | **95.13** | **95.64** |
> | Qwen 1.5b | ✘ | 98.64 | 98.15 | 98.40 | 97.68 | 98.28 | 97.98 | 98.19 |
> | | ✔ | 98.70 | 79.19 | 87.87 | 78.93 | 98.67 | 87.70 | 87.79 |
> | **SCM 1.5b** | **✔** | **98.39** | **97.91** | **98.15** | **97.37** | **97.97** | **97.67** | **97.91** |
> | Qwen 7b | ✘ | 93.87 | 93.29 | 93.58 | 91.57 | 92.28 | 91.93 | 92.75 |
> | | ✔ | 98.09 | 63.18 | 76.85 | 67.87 | 98.44 | 80.34 | 78.60 |
> | **SCM 7b** | **✔** | **97.78** | **97.72** | **97.75** | **97.12** | **97.19** | **97.16** | **97.45** |
>
> # 2 The generalizability of SCM (W2)
>
> ## 2.1 Data diversity in the test set
>
> The sources of harmful responses in FineHarm consist of two parts:
>
> - Harmful responses from the uncensored model based on the harmless data in WildGuard and WildJailbreak.
> - Harmful responses provided by WildGuard, covering 9 LLMs (GPT-4, OLMo-7B-Instruct, GPT-3.5, Vicuna-7b-v1.5, Llama3-8B-Instruct, Mistral-7B-Instruct-v0.2, and 3 Dolphin variants) [1].
>
> Therefore, FineHarm contains harmful responses from multiple LLMs to ensure the source diversity.
>
> ## 2.2 Out-of-distribution (OOD) testing
>
> For OOD setting, we tested SCM (trained on our FineHarm) on two existing, non-overlapping benchmarks, Toxic-Chat [2] and ToxiGen [3], and compared the performance with the OOD baselines derived from [2, 3], i.e., HateBERT and ToxDectRoBERTa.
>
> Toxic-Chat:
>
> | Model | Benign Precision | Benign Recall | Benign F1 | Harmful Precision | Harmful Recall | Harmful F1 | Macro F1 |
> |:--:|:--:|:--:|:--:|:--:|:--:|:--:|:--:|
> | HateBERT | 93.36 | 97.06  | 95.17 | 20.57  | 9.94 | 13.41 | 54.29 |
> | ToxDectRoBERTa| 94.06 | 99.62  | 96.76 | 78.31  | 17.96 | 29.21 | 62.99 |
> | SCM 0.5b | 95.03 | 95.64  | 95.33 | 37.95  | 34.81 | 36.31 | 65.82 |
> | SCM 1.5b | 95.11 | 96.89  | 95.99 | 46.35  | 35.08 | 39.94 | 67.96 |
> | SCM 7b | 95.64 | 96.12  | 95.88 | 45.86  | 42.82 | 44.29 | 70.08 |
>
> ToxiGen:
>
> | Model | Benign Precision | Benign Recall | Benign F1 | Harmful Precision | Harmful Recall | Harmful F1 | Macro F1 |
> |:--:|:--:|:--:|:--:|:--:|:--:|:--:|:--:|
> | HateBERT | 75.97 | 83.23  | 79.43 | 55.60  | 44.37 | 49.36 | 64.39 |
> | ToxDectRoBERTa| 73.42 | 87.46  | 79.83 | 55.56  | 33.11 | 41.49 | 60.66 |
> | SCM 0.5b | 80.11 | 66.93  | 72.93 | 48.16  | 64.90 | 55.29 | 64.11 |
> | SCM 1.5b | 82.83 | 77.90  | 80.29 | 58.53  | 65.89 | 61.99 | 71.14 |
> | SCM 7b | 83.77 | 75.24  | 79.27 | 56.95  | 69.21 | 63.48 | 70.88 |
>
> Under the same OOD setting:
>
> - SCM has a stronger generalization ability than the baselines in the original papers [2,3].
> - On Toxic-chat, the performance of SCM-7B under OOD testing is comparable to the i.i.d. performance reported in [2] (F1=72.1).
>
> > [1] WildGuard: Open one-stop moderation tools for safety risks, jailbreaks, and refusals of LLMs. NeurIPS’24
> >
> > [2] ToxicChat: Unveiling Hidden Challenges of Toxicity Detection in Real-World User-AI Conversation. EMNLP’23 Findings
> >
> > [3] ToxiGen: A Large-Scale Machine-Generated Dataset for Adversarial and Implicit Hate Speech Detection. ACL’22
>
> # 3 Hyperparameter sensitivity analysis of $\beta$ (Q1)
>
> Due to the length limit, the following table shows the impact of $\beta$ on the results of the SCM-1.5B, and it can be seen that the macro F1 generally shows a trend of first rising, then stabilizing, and then fluctuating slightly as the weight of logical constraint changes. When a larger weight is adopted (greater than 1), the penalty for logical inconsistencies tends to be saturated.
>
> | $\beta$ | Benign Precision | Benign Recall | Benign F1 | Harmful Precision | Harmful Recall | Harmful F1 | Macro F1 |
> |:--:|:--:|:--:|:--:|:--:|:--:|:--:|:--:|
> | 0 | 93.30 | 93.53  | 93.42 | 91.79  | 91.50 | 91.65 | 92.53 |
> | 0.2 | 94.21 | 98.28  | 96.20 | 97.69  | 92.36 | 94.95 | 95.58 |
> | 0.4 | 97.33 | 98.65  | 97.98 | 98.26  | 96.57 | 97.41 | 97.69 |
> | 0.6 | 98.09 | 97.91  | 98.00 | 97.36  | 97.58 | 97.47 | 97.73 |
> | 0.8 | 98.33 | 97.97  | 98.15 | 97.44  | 97.90 | 97.67 | 97.91 |
> | 1 | 98.39 | 97.91  | 98.15 | 97.37  | 97.97 | 97.67 | 97.91 |
> | 2 | 98.04 | 98.34  | 98.19 | 97.89  | 97.51 | 97.70 | 97.94 |
> | 5 | 98.50 | 97.17  | 97.83 | 96.48  | 98.13 | 97.30 | 97.56 |
> | 10 | 97.97 | 98.09  | 98.03 | 97.58  | 97.43 | 97.50 | 97.77 |
>
> The other two models (0.5B/7B) show a consistent trend. We will include the results in the subsequent version.
>
> ---
>
> Thank you again for your insightful feedback.
>
> We believe these revisions will strengthen the paper's clarity. Please let us know if further clarifications are needed.
>
> Best regards

---

> > ### Comment · Reviewer_JZGk · 2025-08-06
> >
> > Thanks to the authors' detailed responses and extended results. The author has addressed some of my concerns, and I'll keep my score as it already indicates acceptance.

---

> > > ### Author Response · Authors · 2025-08-07
> > >
> > > Thank you once again for taking the time to review our paper. We will include all experiments mentioned above in the next version.
> > >
> > > Best regards

---

### Comment · Area_Chair_e5xg · 2025-08-05
**Discussion Request**

Hi Reviewers,

If you have not checked the authors' rebuttal, please check and reply. If you do not agree with other reviewers comments, please leave a comment.

Thank you!

AC

---

### Decision · Program_Chairs · 2025-09-17

**Decision:**

Accept (poster)

**Comment:**

This paper focuses on detecting harmful outputs in streaming LLMs. The rebuttal addressed most concerns by adding baselines (like LlamaGuard and APIs), OOD evaluations, ablations, and latency analysis. Some reviewers still see the method as somewhat incremental and note reliance on heuristic labels, but they accept that the experiments are convincing. Overall, this is a solid contribution with good practical impact, and the consensus leans toward acceptance.